# *In vitro* assessment of triterpenoids NVX-207 and betulinyl-bis-sulfamate as a topical treatment for equine skin cancer

Lisa Annabel Weber[1], Anne Funtan[2], Reinhard Paschke[2], Julien Delarocque[1], Jutta Kalbitz[3], Jessica Meißner[4], Karsten Feige[1], Manfred Kietzmann[4], Jessika-Maximiliane V. Cavalleri[5]*

**1** Clinic for Horses, University of Veterinary Medicine Hannover, Foundation, Hannover, Germany, **2** Biozentrum, Martin-Luther-University Halle-Wittenberg, Halle (Saale), Germany, **3** BioSolutions Halle GmbH, Halle (Saale), Germany, **4** Department of Pharmacology, Toxicology and Pharmacy, University of Veterinary Medicine Hannover, Foundation, Hannover, Germany, **5** Equine Internal Medicine, University Equine Clinic, University of Veterinary Medicine Vienna, Vienna, Austria

* Jessika.Cavalleri@vetmeduni.ac.at

**Data Availability Statement:** All relevant data are within the manuscript and its Supporting information files.

## Abstract

Equine sarcoid (ES) is the most prevalent skin tumor in equids worldwide. Additionally, aging grey horses frequently suffer from equine malignant melanoma (EMM). Current local therapies targeting these skin tumors remain challenging. Therefore, more feasible topical treatment options should be considered. In order to develop a topical therapy against ES and EMM, betulinyl-bis-sulfamate and NVX-207, derivatives of the naturally occurring betulin and betulinic acid, respectively, were evaluated for their antiproliferative (crystal violet staining assay), cytotoxic (MTS assay) and apoptotic (AnnexinV staining, cell cycle investigations) effects on primary ES cells, EMM cells and equine dermal fibroblasts *in vitro*. The more potent derivative was assessed for its *in vitr*o penetration and permeation on isolated equine skin within 30 min and 24 h using Franz-type diffusion cells and HPLC analysis. Betulinyl-bis-sulfamate and NVX-207 inhibited the proliferation and metabolism in ES cells, EMM cells and fibroblasts significantly (p < 0.001) in a time- and dose-dependent manner. NVX-207 had superior anticancer effects compared to betulinyl-bis-sulfamate. Both compounds led to the externalization of phosphatidylserines on the cell membrane and DNA fragmentation, demonstrating that the effective mode of action was apoptosis. After 48 h of treatment with NVX-207, the number of necrotic cells was less than 2% in all cell types. Detected amounts of NVX-207 in the different skin layers exceeded the half-maximal inhibitory concentrations calculated by far. Even though data obtained *in vitro* are auspicious, the results are not unconditionally applicable to the clinical situation. Consequently, *in vivo* studies are required to address the antitumoral effects of topically applied NVX-207 in ES and EMM patients.

**Funding:** Some authors were funded by the Central Innovation Programme of the German Federal Ministry for Economic Affairs and Energy. LAW: Specific grant number: TopiDrugHorse 16KN051526 BMWI; AF: Specific grant number: TopiDrugHorse 16KN051530 BMWI; RP: Specific grant number: TopiDrugHorse 16KN051524 BMWI; JK: Specific grant number: TopiDrugHorse 16KN051524 BMWI. RP and JK were further funded by BioSolutions Halle GmbH (www.biosolutions-halle.de). The TopiDrugHorse project is a cooperation project between research institutions and a company. The participation of a company in this cooperation project is the prerequisite for the approval of the project funds by the Ministry of Economic Affairs and Energy. BioSolutions Halle GmbH is therefore not a commercial funder, but an equal partner. The rights and obligations of all partners are governed by a cooperation agreement. This applies in particular to publications and intellectual property. The funders provided support in the form of salaries and/or research materials for authors [LAW, AF, RP, JK], but did not have any additional role in the study design, data collection and analysis, decision to publish, or preparation of the manuscript. The specific roles of these authors are articulated in the 'author contributions' section.

**Competing interests:** The authors declare that no competing interests exist. The affiliation "BioSolutions Halle GmbH" of Dr. Jutta Kalbitz does not alter our adherence to PLOS ONE policies on sharing data and materials. As described in the Funding Statement, BioSolutions Halle GmbH is not a commercial funder, but an equal partner in the TopiDrugHorse project.

## Introduction

The skin is the organ in horses most frequently affected by tumors [1]. With a reported occurrence ranging from 35 to 90% of all cutaneous neoplasms [2–4], the equine sarcoid (ES) is the most prevalent skin cancer in equids worldwide [5–7]. The pathogenesis of this coat-color independent tumor of the fibroblasts has been linked to an infection with the bovine papillomavirus type 1 and 2 [8–10], trauma [11, 12], and a genetic predisposition [13, 14]. According to their gross appearance and clinical behavior, sarcoids are classified into six types: Mild occult or verrucous tumors and more severe nodular, fibroblastic, mixed and malevolent lesions [12]. Even though non-metastasizing and mostly not life-threatening, their locally invasive growth and predilection sites (e.g. head, saddle girth area) can seriously impair the equid's welfare and compromise the use and economic value of the animal [1]. Multiple treatment modalities for the ES are described in the literature (e.g. surgery, radiation, chemotherapy, immunotherapy) but universal effectiveness is not given and recurrence rates are high [11, 15]. Topical therapies generally seem particularly feasible as they are noninvasive and applicable, even on treatment sites that are difficult to access. However, the results regarding the efficacy of the acyclovir cream often used for mild-type ES treatment are contradictory [16, 17] and imiquimod may temporarily cause severe local side effects [18]. In addition, although a variety of other topical treatment options exists, mainly anecdotal evidence of their success is reported [1, 15, 19, 20]. Therefore, the development of a novel topical treatment approach for ES should be considered to take advantage of the benefits of topical therapies.

The equine malignant melanoma (EMM) is a frequently occurring, sex-independent skin neoplasm with a high prevalence in grey horses older than 15 years of age [21–25]. Melanomas are melanocytic tumors which typically occur as nodular in glabrous cutaneous regions (e.g. ventral surface of the tail, perineum, anus, external genitalia) [22, 26]. The dominant age-related phenotype of greying and the predisposition to melanoma are associated with a mutation in intron 6 of the syntaxin-17 gene [27, 28]. Most of the tumors show a slow growth pattern over years, however, more than 60% become malignant and cause clinical problems due to enlargement und widespread metastases [29–31]. Treatment options reported with varying outcomes include systemic and local approaches, such as immunotherapy [32–34], cimetidine application [35, 36], radiation [37], surgery [38, 39], and chemotherapy with cisplatin alone [40, 41] or in combination with electrochemotherapy [42, 43]. Although effective in many cases, surgical excision can be challenging due to the unfavorable localization of the tumors and the intratumoral injection of the mutagenic and carcinogenic cisplatin is linked to strict safety rules [44]. Therefore, a more practical treatment option for early stages of EMM, for example, in the form of a cream, would be useful.

Promising substances for topical ES and EMM treatment could be triterpenoids, such as betulinic acid (BA) and its derivatives [45, 46]. Betulinic acid, the oxidation product of betulin, is a pentacyclic lupane-type triterpenoid and can be extracted from various botanical sources [47]. Since first studies proved BA's antitumor activity against human melanoma and other malignancies in cell culture and animal models [48, 49], a plethora of scientific work has verified the wide range of its biological capabilities *in vitro* and *in vivo* [50, 51]. Treatment with BA induces apoptosis in cancer cells due to a direct effect on the mitochondria [52] independent of CD95 ligand/receptor interaction [49]. Alterations in the mitochondrial membrane potential mediate a cytochrome *c* and apoptosis-inducing factor release, which results in the cleavage of caspases and nuclear disintegration [53, 54]. Furthermore, the generation of reactive oxygen species [49, 55], the subsequent mitogen-activated protein kinase activation [56] and the inhibition of eukaryotic topoisomerase I [57], endothelial-to-mesenchymal-transition [58] and angiogenesis [59, 60] are suggested as BA-mediated antitumoral properties. The anticancer

effects of BA against EMM cells and its potent permeation in isolated equine skin have recently been reported [45]. However, based on a classification for the cytotoxicity of triterpenes [61], the half-maximal inhibitory concentrations (IC$_{50}$) of BA for EMM cells and other human and animal cancer cell lines are considered to be only moderate. In addition, the compounds' hydrosolubility is limited, which reduces the opportunities of medicinal use mainly to topical applications [62]. A variety of synthetically modified derivatives have been synthesized in the past few decades to enhance the pharmacological properties of BA and the closely related compound betulin [62]. Among these are betulinyl-bis-sulfamate ((3β)-Lup-20(29)-ene-3,28-diol, 3,28-disulfamate; BBS) [63] and NVX-207 (3-acetyl-betulinic acid-2-amino-3-hydroxy-2-hydroxymethyl-propanoate) [64], from which, especially the latter substance, shows a higher cytotoxicity in various human and canine cancer cell lines compared to the parent BA [64–66]. It has been demonstrated that NVX-207 induces apoptosis in EMM cells [66]. In addition, the compound has already been successfully tested in a clinical study with canine cancer patients [64]. Within the frame of pilot safety studies, NVX-207 was well tolerated when applied topically in eight healthy horses [67] or injected intralesionally in two horses affected by EMM [66].

The objectives of this study were (1) to investigate the betulin derivative BBS and BA derivative NVX-207 for their antiproliferative, cytotoxic and apoptotic effects on ES cells, EMM cells and equine dermal fibroblasts and (2) to assess the more potent derivative for its penetration and permeation on isolated equine skin *in vitro* with the aim of developing a topical therapy for the ES and EMM.

## Material and methods

### Evaluation of the anticancer effects of BBS and NVX-207 on equine melanoma cells and equine dermal fibroblasts

Compounds. Biosolutions Halle GmbH (Halle/Saale, Germany) synthesized BBS and NVX-207. The compounds were dissolved in dimethyl sulfoxide (WAK-Chemie Medical GmbH, Steinbach, Germany) to achieve 20 mM stock solutions.

**Cells and culture conditions.** All cells used for the experiments originate from different horses. Primary EMM cells (MelDuWi) and primary equine dermal fibroblasts (PriFi1, PriFi2) belong to the cell culture stock of the Clinic for Horses, University of Veterinary Medicine Hannover, Foundation, Hannover, Germany. The cells were cultured as monolayers at 37°C in a humified atmosphere with 5% CO$_2$ and maintained in RPMI1640 cell culture medium with stable glutamine (Biochrom GmbH, Berlin, Germany) supplemented with 15% fetal bovine serum superior (Biochrom GmbH) and 1% penicillin and streptomycin (10,000 international units (I.U.)/mL /10,000 μg/mL, Biochrom GmbH). Primary ES cells sRGO1 and sRGO2 (kindly provided by Dr. Sabine Brandt, University of Veterinary Medicine Vienna, Vienna, Austria) and primary EMM cells eRGO1 (kindly provided by Dr. Barbara Pratscher, University of Veterinary Medicine Vienna, Vienna, Austria) were cultured as monolayers at 37°C in a humified atmosphere with 5% CO$_2$ and kept in Dulbecco's modified Eagle's high glucose w/Glutamax (4.5 g/L) cell culture medium (GIBCO-Invitrogen, Thermofisher, Darmstadt, Germany) supplemented with 10% fetal bovine serum superior (Biochrom GmbH) and 1% Antibiotic-Antimycotic (100x; GIBCO-Invitrogen), containing penicillin (10,000 units/mL), streptomycin (10,000 μg/mL) and amphotericin B (25 μg/mL).

**Proliferation assay.** The proliferation assay was performed as published [45]. Briefly, a modified crystal violet staining assay (CVS) was carried out to investigate the antiproliferative effects of BBS and NVX-207 on primary equine cells. The cells were exposed to BBS and NVX-207 at nine different concentrations ranging from 1–100 μmol/L for 5, 24, 48 and 96 h.

Proliferation and cytotoxicity experiments for this cell type were performed only for 5, 24 and 48 h as even untreated sarcoid cells showed an altered growth behavior in 96 h experiments. Control cells were treated with medium only. The proportion of cells treated relative to untreated controls was determined by crystal violet staining and photometric absorbance measurement at the incubation time points mentioned above. Proliferation assays were performed in six to eight biological replicates with two technical replicates for each combination of cell type, incubation time and compound concentration.

**Cytotoxicity assay.** The cytotoxicity of the compounds was assessed by the CellTiter 96$^{®}$ AQ$_{ueous}$ One Solution Cell Proliferation Assay (MTS) (Promega GmbH, Mannheim, Germany) as reported [45]. In brief, in order to reach cell confluence within 48 h, cells were seeded into 96-well plates in adequate densities (MelDuWi 30,000 cells/well; PriFi1, PriFi2, eRGO1 20,000 cells/well; sRGO1and sRGO2 15,000 cells/well). Incubation times and concentrations of BBS and NVX-207 were applied in accordance with the CVS assay. The formazan dye generated by the metabolic active cells was quantified photometrically. Cytotoxicity assays were performed in six to nine biological replicates with two technical replicates for each combination of cell type, incubation time and compound concentration.

**Cell cycle investigations.** Approximately $7.5 \times 10^5$ cells (MelDuWi) and $1.0 \times 10^6$ cells (PriFri2 and sRGO2) were seeded in 25 cm$^2$ cell culture flasks. After 24 h of incubation, the medium was replaced with medium containing either BBS or NVX-207 at their respective double IC$_{50}$ concentration (measured after 96 h by sulforhodamine B [SRB] assay, analogous to [66]; see S1 and S2 Appendices). Following 24 and 48 h of incubation, the cells were harvested by mild trypsinization and washed twice with phosphate-buffered saline (PBS) buffer (containing Mg$^{2+}$ and Ca$^{2+}$). Cells ($1.0 \times 10^6$) were fixed with ethanol (70%, -20˚C, for 24 h). After discarding the ethanol, the cells were washed in 1 mL PBS buffer (containing Mg$^{2+}$ and Ca$^{2+}$) and were centrifuged. The cell pellet was resuspended in 1 mL of staining PBS buffer (containing Mg$^{2+}$ and Ca$^{2+}$, 10 μg/mL RNASe [Thermofisher] and 15 μg/mL propidium iodide [Sigma-Aldrich, Munich, Germany]) and was incubated for 30 min at room temperature. Analyses were performed using the Attune$^{®}$ FACS machine (Life Technologies, Darmstadt, Germany) collecting data from the BL-2A channel. Doublet cells were excluded from the measurements by plotting BL-2A against BL-2H. A total of 20,000 events were collected for each cell cycle distribution. Each sample was measured in duplicate.

**AnnexinV staining.** Approximately $7.5 \times 10^5$ cells (MelDuWi) and $1.0 \times 10^6$ cells (PriFri2 and sRGO2) were seeded in 25 cm$^2$ cell culture flasks. After 24 h of incubation, the medium was replaced with medium containing either BBS or NVX-207 at their respective double IC$_{50}$ concentration (measured after 96 h). Following 24 and 48 h of incubation, cells were harvested by mild trypsinization and washed twice with PBS buffer (containing Mg$^{2+}$ and Ca$^{2+}$). Cells ($1.0 \times 10^6$) were resuspended in AnnexinV binding buffer (BioLegend$^{®}$, San Diego, US) to a concentration of $1.0 \cdot 10^6$ cells/mL. Approximately 100,000 cells were stained with propidium iodide solution (3 mL, 1 mg/mL) and FITC AnnexinV solution (5 mL, BioLegend$^{®}$) for 15 min in the dark at room temperature. After the addition of Annexin V binding buffer (400 mL), the suspension was analyzed using the Attune$^{®}$ FACS machine (Life Technologies). After gating for living cells, the data from detectors BL-1A and BL-3A were collected. A total of 20,000 events were collected from each sample and technical duplicates were measured.

## Diffusion of NVX-207 into equine skin

**Test formulations.** Two different pharmaceutical formulations were provided by Skinomics GmbH, Halle, Germany, for *in vitro* permeation studies. Based on previous permeation studies with BA [45], test formulation 1 consisted of "Basiscreme DAC" (pharmaceutical

**Table 1. Information about the equine skin donors used for Franz-type diffusion cell experiments.**

| Incubation time | Number of horses | Sex | Breed | Median age in years (range min-max) |
|---|---|---|---|---|
| 30 min | 6 | 3 mares, 2 geldings, 1 unknown | 1 Hanoverian Warmblood, 1 Icelandic horse, 1 Arabian horse, 1 Clydesdale, 2 unknown | 19 (4–23) |
| 24 h | 6 | 2 mares, 4 geldings | 2 Warmblood horses, 1 Hanoverian Warmblood, 1 Holsteiner Warmblood, 1 Arabian horse, 1 Icelandic horse | 16 (6–25) |

amphiphilic cream as published in the German Drug Codex) with 1% NVX-207 and 20% medium-chain triglycerides. The formulation was modified because of an inhomogenous distribution of NVX-207 in test formulation 1 (oily sediments and overall recovery rate < 50% in Franz-type diffusion cells (FDC) experiments): Test formulation 2 contained "Basiscreme DAC" with 1% NVX-207.

**Skin sample preparation and Franz-type diffusion cell experiments.** Skin from six horses was used for each FDC experiment. The skin from the lateral thorax was dissected at the Institute of Pathology, University of Veterinary Medicine Hannover, Foundation, Hannover, Germany, after euthanasia of the horses at the Clinic for Horses, University of Veterinary Medicine Hannover, Foundation, for reasons unrelated to the present study. Therefore, a prospective approval of the experiments by an animal research ethics committee was not required. Skin samples were stored at -20˚C until use (maximum five months). Table 1 provides information about the sex, breed and age of the different equine skin donors. Further skin sample preparation and diffusion experiments were performed as reported [45]. Skin samples were incubated with test formulation 1 for 24 h and with test formulation 2 for 30 min and 24 h, respectively.

**Sample processing and NVX-207 quantification.** Following diffusion experiments, skin sample processing and NVX-207 quantification were performed as published with a few modifications [45]. In short, in order to determine the concentration of NVX-207 in different skin layers, skin samples were cut with a cryostat (CryoStar™ NX70 Cryostat, Thermofisher, Darmstadt, Germany) in slices parallel to the skin surface starting from the epidermal side. The first slice had a thickness of 10 μm and, therefore, included the *stratum corneum* with potential residues of the test formulation, which had not been removed with the cotton swab. The following slices were 20-μm thick. Because of the short incubation time in the experiments (30 min), slices were pooled at 5 × 20 μm to investigate the concentration of NVX-207/100 μm skin depth and, therefore, increase the possibility of finding amounts of NVX-207 above the detection limit (0.1 μg/mL). A higher permeation rate of the compound was expected for 24-h experiments and, therefore, the 20-μm slices were stored and analyzed separately up to a depth of 310 μm. The slices were then pooled at 5 × 20 μm until a depth of a maximal 910 μm was reached. The cryostat blade was cleaned with tissues soaked in 80% methanol between each cut. The quantity of NVX-207 was determined by an analytic high-performance liquid chromatography (HPLC) method. Reverse phase analysis was performed using an Agilent 1100 system (Agilent, Waldbronn, Germany) on a Luna® Omega column (3 μm, PS C18, 100 Å, 150 x 4.6 mm; Phenomenex, Torrance, US) at 30˚C using a gradient method with acetonitrile (0.1% HCOOH)(A):water (0.1% HCOOH)(B) at 1.1 mL/min, (from 60 to 10% B within 7.50 min). The diode array detector was set at 200 nm.

## Statistical analysis

Technical duplicates with a coefficient of variation of more than 20% were excluded from the cell assay analysis. IC$_{50}$ values of BBS and NVX-207 from the proliferation and cytotoxicity

tests were calculated with the pharmacodynamic model 108 of Phoenix® WinNonlin® software (version 8.1, Certara, USA). Additional statistical data analysis was conducted with R 3.5.1. [68]. A generalized additive model was fitted for each test (MTS and CVS) and cell type comparison (primary EMM cells and fibroblasts, and primary ES cells and fibroblasts) using the 'mgcv' package [69]. Compound concentrations and the duration of incubation were modeled as tensor product smooth interacting with compound and cell line. Cell passage was added as a random effect. An appropriate distribution was selected by the visual inspection of residuals. The p-values were obtained by performing a Wald test for each parameter. Statistical significance was set at 0.05.

## Results

### Proliferation inhibition and cytotoxicity of BBS and NVX-207 on equine cells

The antiproliferative and cytotoxic effects of NVX-207 and BBS on ES cells, EMM cells and equine dermal fibroblasts were assessed by the CVS and MTS assay. In general, both compounds had significant inhibitory effects on cell proliferation ($p < 0.001$ in CVS assay for every cell type) and cell viability ($p < 0.001$ in MTS assay for every cell type) compared to untreated controls. However, effects on the cells were dose- and time-dependent. Figs 1 and 2 show the results from the melanoma cell model. Results of the sarcoid cell model are attached as additional files S3 and S4 Appendices. First significant, dose-dependent antiproliferative effects on ES cells, EMM cells and fibroblasts were observed after 24 h of incubation with BBS and after 5 h of incubation with NVX-207. A significant, dose-dependent reduction in cell viability was observed in ES cells, EMM cells and fibroblasts after 5 h of treatment with BBS and NVX-207.

As assessed by determination of $IC_{50}$ values (Table 2) NVX-207 was more active against the investigated equine cells compared to BBS. When the cells were exposed to BBS for 5 h, the quantity of cells affected was too low to calculate the $IC_{50}$ values in both cytotoxicity and proliferation assays. After 48 h, NVX-207 exceeded BBSs' antiproliferative effects about 23 and 29 times in ES cells sRGO1 and sRGO2, respectively, about 25 and 3 times in EMM cells eRGO1 and MelDuWi, respectively, and about 23 and 6 times in fibroblasts PriFi1 and PriFi2, respectively. NVX-207 was about 11 (sRGO1), 25 (sRGO2), 8 (eRGO1), 3 (MelDuWi), 34 (PriFi1) and 9 (PriFi2) times more cytotoxic than BBS.

Selectivity of both compounds towards the different cells varied. Compared to normal fibroblasts, BBS showed a selectivity to both sarcoid and EMM cells in the proliferation assay and a selectivity to eRGO1 and both sarcoid cell types in the cytotoxicity assay. Sarcoid cells were more sensitive to BBS than EMM cells. Normal fibroblasts did not show a better tolerance towards NVX-207 compared to EMM cells; by contrast, MelDuWi were revealed to be more resistant in both assays. A selectivity of NVX-207 towards fibroblasts was observed in the proliferation assay for sarcoid cells.

**Cell cycle investigations.** The cell death mechanisms of NVX-207 and BBS on ES cells, EMM cells and equine dermal fibroblasts were assessed by cell cycle investigations via flow cytometry. Condensation of chromatin and fragmentation of DNA and nuclei occurs in apoptotic cells, which can be detected by the SubG1 peak. In comparison to untreated cells (control), the treatment with BBS and NVX-207 caused an increase of subG1 cells after 48 h of treatment for all equine cells (Fig 3 and S5–S12 Appendices). The subG1 peak for the EMM cells MelDuWi arose after a treatment of 48 h to more than 40% for BBS and more than 60% for NVX-207. The equine dermal fibroblasts PriFi2 also showed an increased numbers of subG1 cells (> 80%) after 48 h of treatment with NVX-207 but only 14% after 48 h of

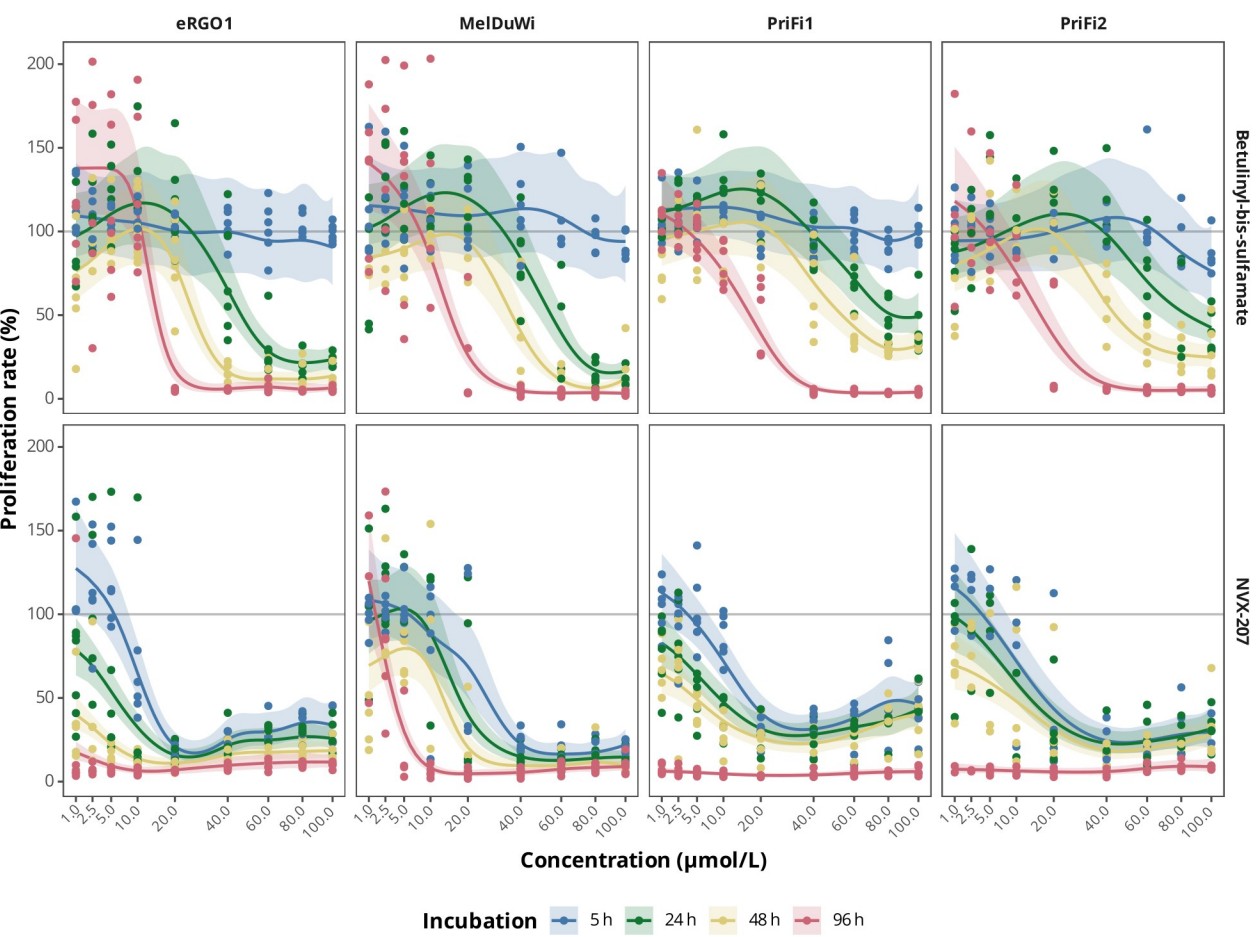

**Fig 1. Effects of betulinyl-bis-sulfamate and NVX-207 on primary equine cell proliferation at different time points.** Proliferation dose-response regression lines of betulinyl-bis-sulfamate (BBS) and NVX-207 on primary equine malignant melanoma (EMM) cells (eRGO1, MelDuWi) and primary equine dermal fibroblasts (PriFi1, PriFi2) at four different time points (5, 24, 48 and 96 h) determined by CVS assay. Antiproliferative effects of the compounds on primary equine cells increase with the concentration and time of drug exposition. Data represent regression lines and 95% confidence intervals of 6–8 independent experiments for each combination of cell type, incubation time and concentration. Concentrations at which the corresponding 95% confidence intervals do not cross the 100% line indicate a significant reduction of the proliferation rate.

treatment with BBS (Fig 4 and S12 Appendix). Thus, a selectivity of BBS for the initiation of the preferably programmed cell death in EMM cells could be shown. The effect of both active substances on the sarcoid cells was noticeably lower compared to the other cell lines. After a treatment time of 48 h, an enrichment of 20% subG1 cells was present.

**AnnexinV staining.** The externalization of phosphatidylserines to the extracellular side of the plasma membrane is a characteristic and early event in apoptosis [70, 71]. The change of the extracellular plasma membrane composition was detected by using AnnexinV-FITC/ (propidium iodide) staining and analysis by flow cytometry (Figs 5 and 6 and S13–S20 Appendices). Untreated cells were used for control. After a treatment period of 24 h with BBS, 19% of the sarcoid cells were early apoptotic and 39% were late apoptotic, while 2% of the control cells were early and 14% were late apoptotic. After 48 h, the number of apoptotic cells further increased and approximately 90% of the cells were apoptotic (8% early apoptotic; 82% late apoptotic). NVX-207 had a weaker effect on the sarcoid cells and 40% were present as living cells after 48 h. The equine dermal fibroblasts showed a slower increase of apoptotic cells after 24 h of treatment with BBS. In this case, increases of 5% early apoptotic and 3% late apoptotic cells

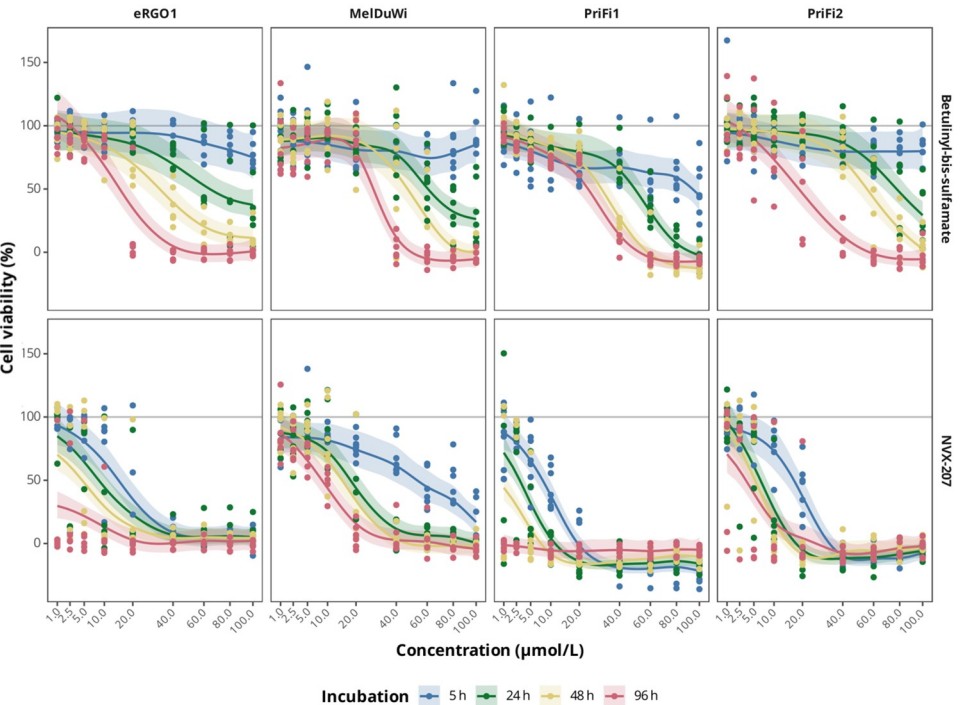

**Fig 2. Effects of betulinyl-bis-sulfamate and NVX-207 on primary equine cell viability at different time points.**
Cytotoxicity dose-response regression lines of BBS and NVX-207 on primary EMM cells (eRGO1, MelDuWi) and
primary equine dermal fibroblasts (PriFi1, PriFi2) at four different time points (5, 24, 48 and 96 h) determined by MTS
assay. Cytotoxic effects of the compounds on primary equine cells increase with concentration and time of drug
exposition. Data represent regression lines and 95% confidence intervals of 6–9 independent experiments for each
combination of cell type, incubation time and concentration. Concentrations at which the corresponding 95%
confidence intervals do not cross the 100% line indicate a significant reduction of cell viability.

were present. However, an increase in late apoptotic cells after 48 h was observed (54% by
treatment with BBS and 67% with NVX-207). Thus, it was shown that BBS had a better selec-
tivity to sarcoid cells compared to fibroblasts.

Compared to BBS, NVX-207 had the stronger potential to induce apoptosis in EMM cells.
After 48 h, 45% were late apoptotic cells and only 30% were living cells. In addition, 25% of
cells were present in the early apoptotic phase. It was proven for all three equine cell lines that
the necrosis rate after 48 h of treatment with NVX-207 was below 2%.

## Diffusion of NVX-207 into equine skin and overall NVX-207 recoveries

When the skin samples were treated with test formulation 2 for 30 min, NVX-207 was detected
in both the epidermis and dermis (Fig 7). An incubation time of 24 h led to an accumulation
of the compound in the upper epidermis (11–30 μm) but did not increase the amount of
NVX-207 in the other skin layers analyzed (Fig 7 and S21 Appendix). The detected concentra-
tions exceeded the 24 h $IC_{50}$ values of NVX-207 for ES cells, EMM cells and equine dermal
fibroblasts determined in the proliferation and cytotoxicity assays even in the deeper skin lay-
ers examined (up to a depth of 810 μm). The overall NVX-207 recovery rate after 30 min of
incubation was 89 ± 23% (mean ± SD; n = 6), from which 68 ± 18% of the substance was
detected in the non-permeated proportion (cotton swabs) and 28 ± 17% in the skin. The over-
all recovery rate of NVX-207 in test formulation 2 after 24 h of incubation was 85 ± 14%
(mean ± SD; n = 6). A quantity of 51 ± 9% of the NVX-207 amount applied was found in the

**Table 2. IC$_{50}$ values (μmol/L) of betulinyl-bis-sulfamate (BBS) and NVX-207 for primary equine cells determined by CVS and MTS assay after 5, 24, 48 and 96 h of drug exposure.**

| | 5 h | | | |
|---|---|---|---|---|
| | Compound and assay | | | |
| | BBS | | NVX-207 | |
| Cells | CVS | MTS | CVS | MTS |
| sRGO1 | - | - | 7 (5–10) | 9 (7–11) |
| sRGO2 | - | - | 6 (4–8) | 8 (6–11) |
| eRGO1 | - | - | 10 (7–13) | 9 (4–15) |
| MelDuWi | - | - | 20 (13–26) | - |
| PriFi1 | - | - | 11 (-2–23) | 11 (9–13) |
| PriFi2 | - | - | 14 (4–24) | 20 (18–22) |
| | **24 h** | | | |
| | Compound and assay | | | |
| | BBS | | NVX-207 | |
| Cells | CVS | MTS | CVS | MTS |
| sRGO1 | 40 (31–49) | - | 7 (5–10) | 4 (2–5) |
| sRGO2 | 38 (33–43) | 45 (40–49) | < 1 (0–2) | 3 (2–4) |
| eRGO1 | 42 (36–48) | 47 (37–57) | 5 (3–7) | 7 (4–15) |
| MelDuWi | 50 (38–61) | 60 (30–91) | 16 (11–21) | 18 (15–21) |
| PriFi1 | 52 (41–62) | 59 (50–68) | 4 (2–6) | 4 (2–5) |
| PriFi2 | 62 (48–76) | 77 (35–118) | 8 (4–12) | 7 (5–9) |
| | **48 h** | | | |
| | Compound and assay | | | |
| | BBS | | NVX-207 | |
| Cells | CVS | MTS | CVS | MTS |
| sRGO1 | 23 (16–31) | 25 (21–30) | < 1 (0–1) | 2 (1–4) |
| sRGO2 | 29 (19–31) | 28 (21–35) | < 1 (< 0–8) | 1 (< 1–2) |
| eRGO1 | 25 (7–44) | 32 (26–38) | < 1 (< 0 –< 1) | 4 (1–7) |
| MelDuWi | 36 (26–46) | 53 (41–65) | 12 (6–18) | 15 (12–19) |
| PriFi1 | 42 (32–51) | 35 (31–39) | 2 (1–3) | 1 (< 1–2) |
| PriFi2 | 39 (32–46) | 61 (48–74) | 7 (< 0–15) | 7 (< 1–7) |
| | **96 h** | | | |
| | Compound and assay | | | |
| | BBS | | NVX-207 | |
| Cells | CVS | MTS | CVS | MTS |
| sRGO1 | n.a. | n.a. | n.a. | n.a. |
| sRGO2 | n.a. | n.a. | n.a. | n.a. |
| eRGO1 | 15 (5–25) 0.04 | 16 (13–18) | < 1 (< 0 – | < 1 (< 0 –< 1) |
| MelDuWi | 16 (4–29) | 32 (15–49) | 4 (3–5) | 8 (5–10) |
| PriFi1 | 17 (15–20) | 28 (26–31) | < 1 (< 0 –< 1) | < 1 (< 0 –< 1) |
| PriFi2 | 16 (11–21) | 20 (11–28) | < 1 (< 0 –< 1) | 4 (< 1–7) |

Antiproliferative (CVS assay) and cytotoxic (MTS assay) effects of BBS and NVX-207 on primary ES cells (sRGO1 and sRGO2), primary EMM cells (eRGO1 and MelDuWi) and primary equine dermal fibroblasts (PriFi1 and PriFi2) after a treatment duration of 5, 24, 48 or 96 h. Data represent mean IC$_{50}$ values (μmol/L) of 6–9 independent experiments with 95% confidence interval in parentheses. "-" = Quantity of cells affected was too low to calculate IC$_{50}$ values with the software applied; "n.a." = data not available

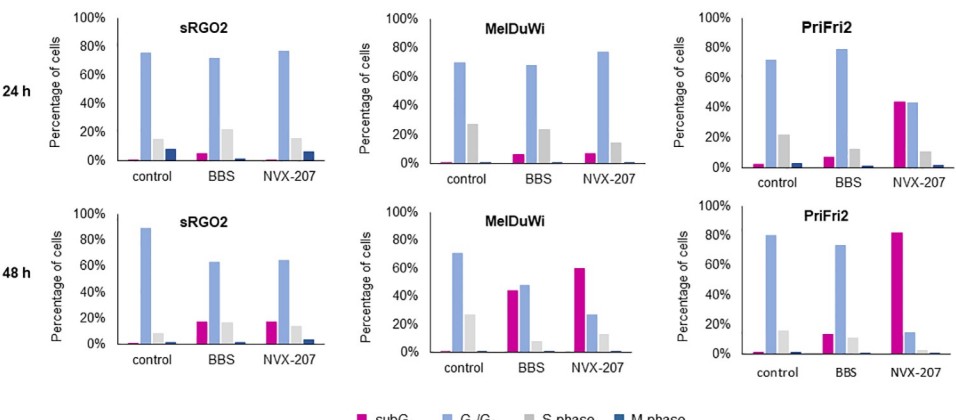

**Fig 3. Cell cycle distributions of sRGO2, MelDuWi and PriFi2.** Percentage of the four different phases in the cell cycle investigation of equine sarcoid (ES) cells (left), EMM cells (middle) and equine dermal fibroblasts (right) treated with BBS and NVX-207 at their double $IC_{50}$ concentrations for 24 and 48 h. Magenta: subG1; light blue: G1/G0; grey: S-phase; and dark blue: G2/M-phase.

cotton swabs and 32 ± 12% of the NVX-207 amount applied was detected in the skin. No NVX-207 was detected in the acceptor medium in any of the FDC experiments.

## Discussion

The ES is the dermatologic neoplasm in equids diagnosed most frequently. The EMM is also a common skin tumor, especially in aging grey horses. In order to develop a topical therapy

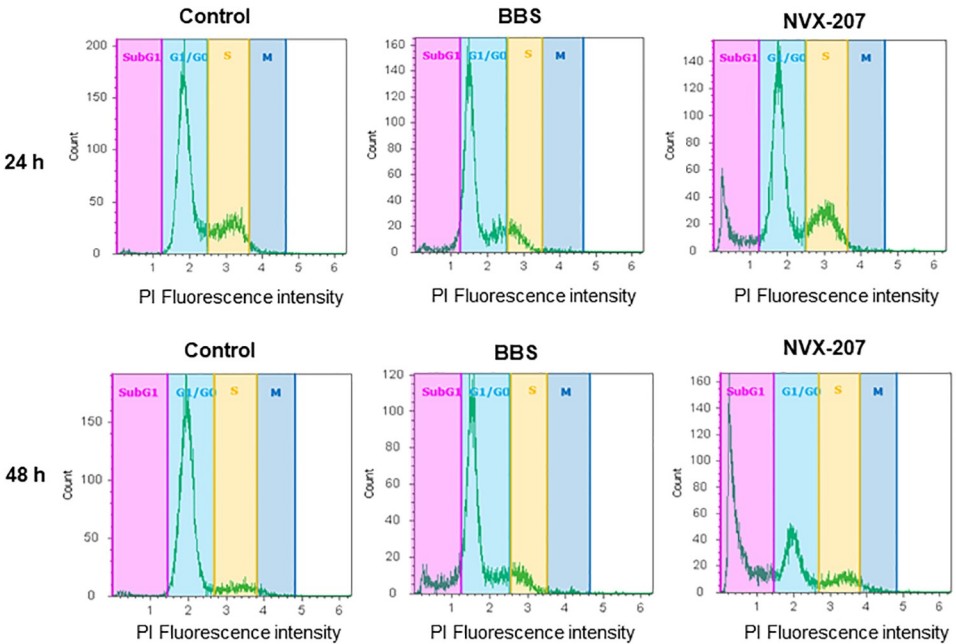

**Fig 4. Cell cycle distributions for equine dermal fibroblasts PriFri2.** The cells were untreated (control) or treated with BBS and NVX-207 at their double $IC_{50}$ concentrations for 24 and 48 h (as indicated). The DNA was stained with propidium iodide and the cells were analyzed by flow cytometry. Red: SubG1 peak; light blue: G1/G0-phase peak; yellow: S-phase peak; and dark blue: G2/M-phase. (See S5–S10 Appendices for the interpretation of the other cell lines).

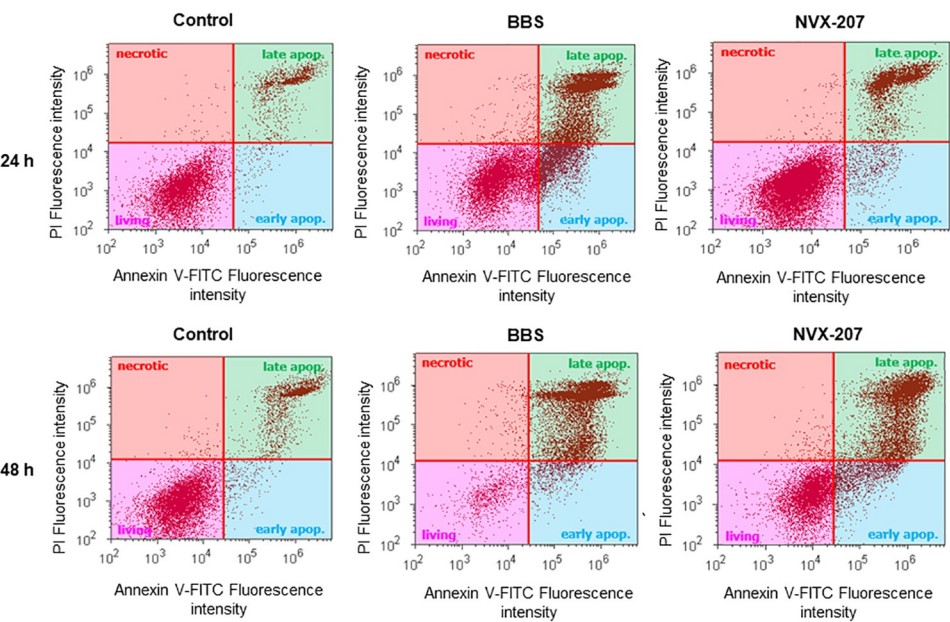

**Fig 5. AnnexinV staining of ES cells sRGO2.** The cells were untreated (control) or treated with BBS and NVX-207 at their double $IC_{50}$ concentrations for 24 and 48 h (as indicated). After harvesting, the cells were stained and flow cytometry analysis was performed. Red: necrotic cells; green: late apoptotic cells; blue: early apoptotic cells; and magenta: living cells.

against the ES and EMM, the betulinic acid derivative NVX-207 and the betulin derivative BBS were assessed for their antiproliferative, cytotoxic and apoptotic effects on ES cells, EMM cells and fibroblasts *in vitro*. Both substances had significant anticancer effects on the cells and induced apoptosis. NVX-207 was revealed to be the more potent substance. Therefore, this

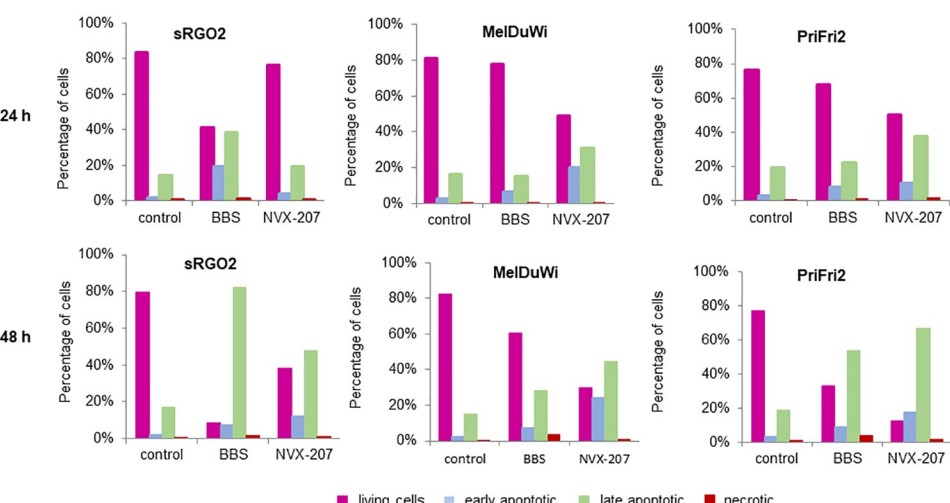

**Fig 6. AnnexinV staining of equine cells.** Equine sarcoid cells sRGO2, EMM cells MelDuWi and equine dermal fibroblasts PriFi2 were untreated (control) or treated with BBS and NVX-207 at their double $IC_{50}$ concentrations for 24 and 48 h (as indicated) and used for the AnnexinV assay. Data shown are the percentages of living cells (magenta), early apoptotic cells (light blue), late apoptotic cells (green) and necrotic cells (red).

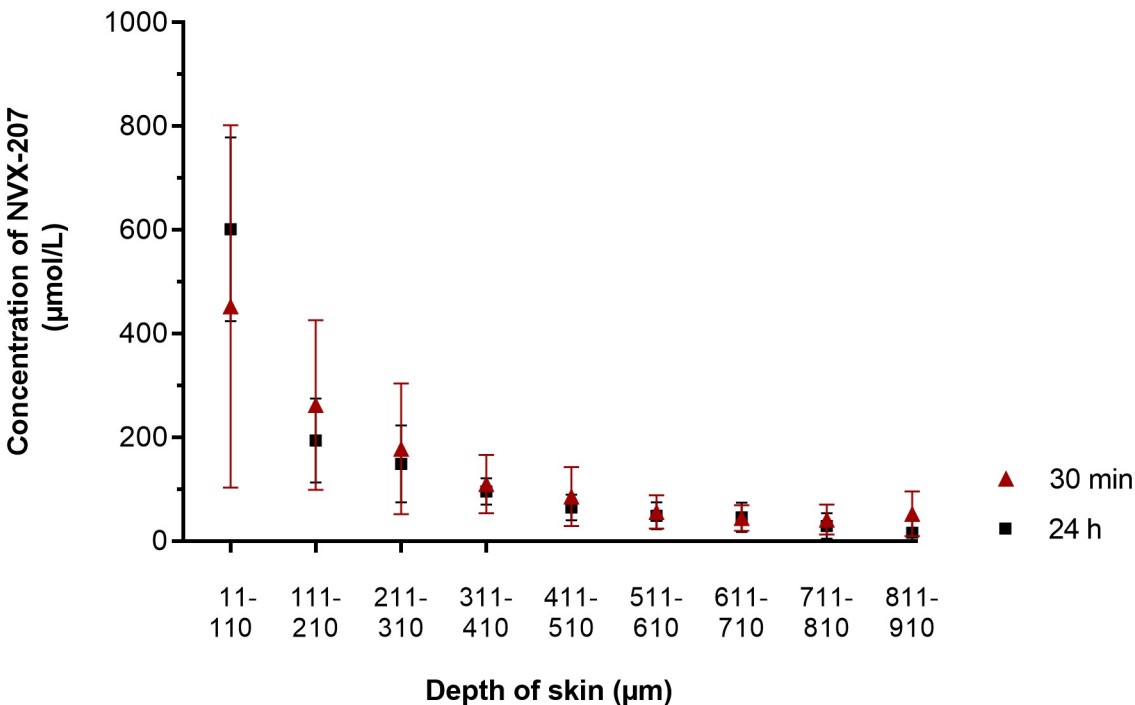

**Fig 7. Concentration profile of NVX-207 correlative to skin thickness after 30 minutes and 24 hours of incubation.** The skin of six horses (two technical replicates each) for each Franz-type diffusion cell experiment was used to investigate the permeation of 1% NVX-207 in "Basiscreme DAC" within 30 min and 24 h. The concentration of the compound for 30-min experiments was determined in 100 μm cryostat skin slices (pooled at 5 × 20 μm) at different skin depths by HPLC analysis. The 20-μm slices for 24-h experiments were stored and analyzed separately up to a depth of 310 μm. The slices were then pooled at 5 x 20 μm until a depth of a maximal of 910 μm was reached. All data are given per 100 μm skin depth in this figure for better comparison of the 30-min and 24-h concentration profiles. A more detailed version of the 24-h permeation profile is provided in the supplemented appendix (S21 Appendix). An incubation time of 24 h led to an accumulation of the compound in the upper epidermis (11–30 μm) but did not increase the amount of NVX-207 in the other skin layers analyzed. Figure data represent the mean concentration of NVX-207 at the skin depths indicated ± SD. Data for the 10-μm skin depth (*stratum corneum*) with potential test compound residues were excluded in this figure.

compound was used for further *in vitro* permeation studies, where it was demonstrated that high concentrations could be reached in isolated equine skin.

The compound NVX-207 was previously assessed for its cytotoxic effects on EMM cells "MelDuWi" with the sulforhodamine B assay and a 96-h $IC_{50}$ value of 5.6 μmol/L was reported [66]. Results of this first study on EMM cells "MelDuWi" could be replicated in the present study with different methodological approaches (CVS and MTS assay to assess the antiproliferative and cytotoxic effects, respectively) and widened by investigations with EMM cells "eRGO1," ES cells "sRGO1" and "sRGO2," and equine dermal fibroblasts "PriFi1" and "PriFi2." Three further treatment time points (5, 24 and 48 h) were included in the experiments to provide more information about the time-dependent efficacy of the drug. It was demonstrated that the antiproliferative and cytotoxic effects on ES cells, EMM cells and dermal fibroblasts enhanced with an increased treatment duration in a dose-dependent manner. After 48 and 96 h, very low concentrations of NVX-207 were sufficient to observe an inhibitory effect on the cells' proliferation and survival rate (e.g. for EMM cells eRGO1 < 1 μmol/L in CVS and MTS assay after 96 h of incubation). In addition, after 5 h of drug exposure, the quantity of affected cells was already high enough to calculate $IC_{50}$ values, substantiating the potent effects of NVX-207 on equine cells. These data could be taken into account when prospective *in vivo* treatment regimens are designed.

This study is the first report on the influence of NVX-207 on ES cells and normal equine dermal fibroblasts. NVX-207 had cell viability reducing and antiproliferative effects on both cell types. The similar treatment response of the cells is not surprising, as the ES is addressed as a tumor of the fibroblasts [12]. However, compared to normal equine cells, a selectivity of the compound to ES cells could be demonstrated in the proliferation assay, suggesting that sarcoid cells are even more sensitive. A selectivity of NVX-207 to EMM cells was not observed. The same was shown for EMM cells and fibroblasts when treated with the parent BA [45]. In contrast to these findings, it was reported that NVX-207 had little impact on the *in vitro* survival of normal human umbilical vein endothelial cells, fibroblasts and keratinocytes [64]. Furthermore, current *in vivo* data indicate a good systemic and local tolerability of 1% NVX-207 after topical application twice a day for seven consecutive days in eight healthy horses [67]. In addition, the intralesional injection of the compound in two EMM patients once a week for 19 consecutive weeks proved to be safe [66]. Intravenous application of the compound in mice did not lead to any side effects [64] and the intralesional treatment of different malignancies in five dogs was well tolerated and clinically beneficial tumor response was observed [64].

It has been demonstrated previously that NVX-207 triggers the mitochondrial-induced apoptotic pathway in human melanoma cell lines via activation of caspases-9, -3 and -7 and cleavage of poly (ADP-ribose) polymerase [64]. Furthermore, an increase of subG1 cells after treatment of various human cancer cell lines with BA and NVX-207 has been reported [72, 73]. An induction of both initiator caspases (caspase-8 and caspase-9) in EMM cells led to an activation of effector caspase-3 [66]. Comparable to a treatment with the parent BA, an accumulation of EMM cells in the subG1 phase and externalization of phosphatidylserines to the extracellular side of the plasma membrane, a characteristic feature of apoptosis, were observed after treatment with NVX-207 [66]. These preliminary investigations by Liebscher et al. on EMM cells MelDuWi could be reproduced in this study. However, up to now, no data on the molecular mechanisms in ES cells and normal equine cells after treatment with NVX-207 has existed. Cell cycle investigations and AnnexinV staining were performed to address this lack. Results from these apoptosis tests demonstrated that NVX-207 triggers apoptosis in ES cells sRGO2. However, the effects were less pronounced compared to EMM cells MelDuWi and equine dermal fibroblasts PriFi2. After 48 h, the number of apoptotic cells detected with AnnexinV staining was about 60%, of which 48% were late apoptotic. Only 17% of the cells were found to be in the subG1 phase with a fragmented DNA. The different quantity of apoptotic cells analyzed with different methods may be explained by the temporally staggered occurrence of characteristic cellular changes, which are made visible by the respective method. The results reported here further indicate that similar modes of action observed in equine cancer cells also take place in unaltered equine cells when treated with NVX-207. After an incubation of 48 h, about 85% of equine dermal fibroblasts PriFi2 were apoptotic and a clear shift to cells in the subG1 phase was already observed after 24 h of treatment. It is remarkable that the proportion of necrotic cells, whether in altered or normal cell types, was below 2% after a treatment of 48 h. Even though the results reported from *in vitro* experiments with equine skin cancer cells are promising, it must be emphasized that cells in their native microenvironment can be much more robust against (phyto)chemotherapeutic influences [74–76]. Therefore, no reliable conclusions can be drawn regarding the efficacy of a topical NVX-207 application in ES and EMM patients and prospective *in vivo* studies have made to address this question.

The betulin derivative BBS had significant antiproliferative and cytotoxic effects on all three cell types investigated in the present study, however, it was considerably less effective compared to NVX-207. In addition, the $IC_{50}$ values of BBS calculated for EMM cells were higher than the ones reported for BA [45, 66]. Therefore, further permeation studies were performed with NVX-207. Nevertheless, in contrast to NVX-207, the compound was less toxic for normal

cells. In order to clarify the cellular pathways of BBS in ES cells, EMM cells and equine dermal fibroblasts, it was shown by AnnexinV staining and cell cycle investigations that BBS induced apoptosis in these cells. However, while the apoptotic impact of BBS was stronger in sarcoid cells compared to the effects of NVX-207 in these equine skin cancer cells, this was not the case for EMM cells and fibroblasts. In EMM cells, the amount of late apoptotic cells after 48 h was 44.8% (NVX-207) compared to 28.2% (BBS). Regarding the results from the proliferation test and the cell cycle analysis, there seems to be a selectivity of BBS towards equine skin cancer cells in comparison to unaltered equine dermal fibroblasts. In addition, AnnexinV staining revealed a preferred triggering of the programmed cell death for the sarcoid cells (82.1% late apoptotic after a 48-h treatment with BBS) when compared to the late apoptotic phase of fibroblasts (53.6%). However, only 28.2% of EMM cells were late apoptotic at this stage.

In addition to its apoptotic effects, it should be noted that BBS has been demonstrated to be an efficient inhibitor of human carbonic anhydrase isoenzymes I, II and IX [63]. Carbonic anhydrase IX is overexpressed in many tumors and involved in complex pathways leading to changes in the tumor microenvironment and subsequent tumor progression [77]. Human malignant melanoma cells also express this enzyme and a combination of proton pump and carbonic anhydrase IX inhibitors led to enhanced anticancer effects in these cells *in vitro* [78]. Further investigations are necessary to confirm and expand these results in equine malignancies, however, carbonic anhydrase inhibitors such as BBS could represent potential candidates as anti-tumor agents alone or adjunctive therapeutic drugs.

Except for ulcerated tumors, histopathologic examinations address the localization of melanocytic skin tumors in horses mostly as "dermal" or "subcutaneous" [79, 80]. The ES is regarded as a neoplasm of the dermal fibroblasts, which appear with an increased density and proliferation [81, 82]. Epidermal alterations, such as hyperplasia, hyperkeratosis or rete pig formation, vary between the different clinical ES types but are present in the majority of cases [82]. Due to the tumors' microscopic appearance, the topically applied compound NVX-207 needs to liberate from the drug formulation, penetrate the body protective *stratum corneum* and permeate through the viable epidermal and dermal strata to reach the sarcoid and melanoma cells. A standardized use of ES or EMM skin was not possible for FDC experiments due to technical reasons, which is a limitation of the study. Therefore, normal thoracic equine skin was utilized, as described previously [45, 83].

It has been reported previously that high concentrations of BA could be reached in isolated equine skin when 1% of the compound was mixed in "Basiscreme DAC" with 20% medium-chain triglycerides [45]. Therefore, a drug formulation containing "Basiscreme DAC" with 20% medium-chain triglycerides and 1% of betulinic acid derivative NVX-207 (test formulation 1) was initially tested for *in vitro* permeation. A significant phase separation was already observed 24 h after the production of test formulation 1. The oily sediments were probably the 20% medium-chain triglycerides added, which coalesced as the emulsifier system combined with 1% NVX-207 was presumably not strong enough to form a stable emulsion with the additional fatty acids. The inhomogeneous distribution of NVX-207 suspected in test formulation 1 was confirmed when less than 50% of the substance, which had allegedly been applied on the diffusion area, was detected in the HPLC analysis. The drug formulation was improved as such a low recovery rate in permeation studies and such high variations of active compound distribution in the cream are not acceptable for a topical medication. When 1% NVX-207 was mixed with "Basiscreme DAC" but without additional medium-chain triglycerides (test formulation 2), no phase separation was observed by visual inspection and the overall recovery rate was above 85%.

There was a nearly identical concentration profile of the compound in isolated equine skin when incubated for 30 min and 24 h, except for a considerable difference in the upper

epidermal layers. This indicates a rapid penetration of the lipophilic NVX-207 through the *stratum corneum* and accumulation in the viable epidermal skin layers, followed by a slower permeation into the subjacent, more hydrophilic dermal skin layers [84, 85]. As the blood circulation in *in vitro* FDC experiments is missing, no compound is absorbed by dermal capillary blood vessels, which could further explain the steady state between the 30-min and 24-h permeation studies. Regarding the *in vitro* data determined about antiproliferative and cytotoxic effects of NVX-207 towards ES cells and EMM cells reported here and formerly for EMM cells [66], the concentrations of the compound reached up to a depth of 810 μm in isolated equine skin after 30 min and 24 h of incubation *in vitro* would be sufficient to have an inhibitory or even cytotoxic impact on the cells' metabolism. This might suggest that the proliferation and survival rate of ES and EMM cells especially in the superficial dermal skin layer could be reduced by NVX-207 *in vivo*. However, as mentioned previously, the epidermal nature in ES varies and epidermal thickening can influence the permeation rate of a topically applied drug negatively [86]. Furthermore, it should be considered that the *in vitro* permeation of acyclovir in ES skin differs significantly from epidermal to superficial dermal and deep dermal skin layers and that less acyclovir was found in the deep dermal layers of sarcoid skin compared to normal skin [87]. By contrast, the *in vitro* concentration profiles of NVX-207 in thoracic skin and hairless EMM predilection site skin (e.g. undersurface on the tail, perianal region) can be assumed to be comparable, as the concentrations of hydrocortisone, a lipophilic substance similar to NVX-207, did not differ significantly in the clipped thoracic equine skin and nearly glabrous groin skin [88]. However, an increased vascularization was described in some EMM [31, 79]. Compound elimination by dermal blood vessels cannot be evaluated by FDC experiments and, therefore, the permeated dose required to exert antitumoral effects *in vivo* can also be significantly higher. Furthermore, an encapsulation of the tumor could reduce the drug permeation rate at the treatment site. Because the *in vitro* anticancer effects were demonstrated to be concentration- and time-dependent, prospective *in vivo* treatment regimens with short application intervals and long treatment durations could favorably influence the concentration and efficacy of NVX-207 in the skin of ES and EMM patients.

## Conclusion

In conclusion, the betulinic acid derivative NVX-207 has a superior antiproliferative and cell viability reducing effect on primary ES cells and EMM cells compared to BBS. Both compounds induced apoptosis. High concentrations of NVX-207 were reached in isolated equine skin–even after only 30 min of incubation–demonstrating a potent skin permeation. Although the *in vitro* data reported are promising, the results are not unconditionally applicable to the clinical situation. Therefore, *in vivo* studies are needed to assess the antitumoral effects of topically applied NVX-207 in equine patients suffering from ES or EMM.

## Supporting information

**S1 Appendix. IC$_{50}$ values measured by SRB Assay after 96 h.** IC$_{50}$ values (μmol/L) of betulinyl-bis-sulfamate (BBS) and NVX-207 thereof on three equine cell types (equine sarcoid [ES] cells sRGO2, equine malignant melanoma [EMM] cells MelDuWi and equine dermal fibroblasts PriFi2) determined by SRB-Assay after 96 h of drug exposure. Measurements were carried out at least as thrice determination.
(DOCX)

**S2 Appendix. Cytotoxicity dose-response curves of BBS and NVX-207.** ES cells sRGO2 (left), EMM (middle) and equine fibroblasts PriFri2 (right) determined by SRB Assay after 96

h (one representative of three independent experiments).
(PNG)

**S3 Appendix. Effects of BBS and NVX-207 on primary equine cell proliferation at different time points.** Proliferation dose-response regression lines of BBS and NVX-207 on primary ES cells (sRGO1, sRGO2) and primary equine dermal fibroblasts (PriFi1, PriFi2) at three different time points (5, 24 and 48 h) determined by crystal violet staining assay. Antiproliferative effects of the compounds on primary equine cells increase with concentration and time of drug exposition. Data represent regression lines and 95% confidence intervals of 6–8 independent experiments for each combination of cell type, incubation time and concentration. Concentrations at which the corresponding 95% confidence intervals do not cross the 100% line indicate a significant reduction of the proliferation rate.
(PNG)

**S4 Appendix. Effects of BBS and NVX-207 on primary equine cell viability at different time points.** Proliferation dose-response regression lines of BBS and NVX-207 on primary ES cells (sRGO1, sRGO2) and primary equine dermal fibroblasts (PriFi1, PriFi2) at three different time points (5, 24 and 48 h) determined by MTS assay. Cytotoxic effects of the compounds on primary equine cells increase with concentration and time of drug exposition. Data represent regression lines and 95% confidence intervals of 6–8 independent experiments for each combination of cell type, incubation time and concentration. Concentrations at which the corresponding 95% confidence intervals do not cross the 100% line indicate a significant reduction of the cell viability rate.
(PNG)

**S5 Appendix. Cell cycle distributions of ES cells sRGO2.** Cells were untreated (control) or treated with BBS and NVX-207 at their double $IC_{50}$ concentrations for 24 and 48 h (as indicated). The DNA was stained with propidium iodide and the cells were analyzed by flow cytometry. Red: SubG1 peak; light blue: G1/G0 phase peak; Yellow: S-phase peak; and dark blue: G2/M phase.
(PNG)

**S6 Appendix. Cell cycle percentage of ES cells sRGO2.** Cells were untreated (control) or treated with BBS and NVX-207 at their double $IC_{50}$ concentrations for 24 h.
(DOCX)

**S7 Appendix. Cell cycle percentage of ES cells sRGO2.** Cell were untreated (control) or treated with BBS and NVX-207 at their double $IC_{50}$ concentrations for 48 h.
(DOCX)

**S8 Appendix. Cell cycle distributions of EMM cells MelDuWi.** Cells were untreated (control) or treated with BBS and NVX-207 at their double $IC_{50}$ concentrations for 24 and 48 h (as indicated). The DNA was stained with propidium iodide and the cells were analyzed by flow cytometry. Red: SubG1 peak; light blue: G1/G0 phase peak; Yellow: S-phase peak; and dark blue: G2/M phase.
(PNG)

**S9 Appendix. Cell cycle percentage of EMM MelDuWi.** Cells were untreated (control) or treated with BBS and NVX-207 at their double $IC_{50}$ concentrations for 24 h.
(DOCX)

**S10 Appendix. Cell cycle percentage of EMM MelDuWi.** Cells were untreated (control) or treated with BBS and NVX-207 at their double $IC_{50}$ concentrations for 48 h.
(DOCX)

**S11 Appendix. Cell cycle percentage of equine dermal fibroblasts PriFri2.** Cells were untreated (control) or treated with BBS and NVX-207 at their double $IC_{50}$ concentrations for 24 h.
(DOCX)

**S12 Appendix. Cell cycle percentage of equine dermal fibroblasts PriFri2.** Cells were untreated (control) or treated with BBS and NVX-207 at their double $IC_{50}$ concentrations for 48 h.
(DOCX)

**S13 Appendix. AnnexinV staining.** Percentage of ES cells sRGO2 untreated (control) or treated with BBS and NVX-207 at their double $IC_{50}$ concentrations for 24 h.
(DOCX)

**S14 Appendix. AnnexinV staining.** Percentage of ES cells sRGO2 untreated (control) or treated with BBS and NVX-207 at their double $IC_{50}$ concentrations for 48 h.
(DOCX)

**S15 Appendix. AnnexinV staining of equine dermal fibroblasts PriFri2.** Cells were untreated (control) or treated with BBS and NVX-207 at their double $IC_{50}$ concentrations for 24 and 48 h (as indicated). After harvesting, the cells were stained and flow cytometry analysis was performed. Red: necrotic cells; green: late apoptotic cells; blue: early apoptotic cells; magenta: living cells.
(PNG)

**S16 Appendix. AnnexinV staining.** Percentage of equine dermal fibroblasts PriFri2 untreated (control) or treated with BBS and NVX-207 at their double $IC_{50}$ concentrations for 24 h.
(DOCX)

**S17 Appendix. AnnexinV staining.** Percentage of equine dermal fibroblasts PriFri2 untreated (control) or treated with BBS and NVX-207 at their double $IC_{50}$ concentrations for 48 h.
(DOCX)

**S18 Appendix. AnnexinV staining of EMM cells MelDuWi.** Cells were untreated (control) or treated with BBS and NVX-207 at their double IC50 concentrations for 24 and 48 h (as indicated). After harvesting, the cells were stained and flow cytometry analysis was performed. Red: necrotic cells; green: late apoptotic cells; blue: early apoptotic cells; magenta: living cells.
(PNG)

**S19 Appendix. AnnexinV staining.** Percentage of EMM cells (MelDuWi) untreated (control) or treated with BBS and NVX-207 at their double $IC_{50}$ concentrations for 24 h.
(DOCX)

**S20 Appendix. AnnexinV staining.** Percentage of EMM cells (MelDuWi) untreated (control) or treated with BBS and NVX-207 at their double $IC_{50}$ concentrations for 48 h.
(DOCX)

**S21 Appendix. Concentration profile of NVX-207 correlative to skin thickness after 24 h of incubation.** The skin of six horses (two technical replicates each) were used to investigate the permeation of 1% NVX-207 in "Basiscreme DAC" within 24 h for the Franz-type diffusion cell experiment. The concentration of the compound was determined in 20 μm and 100 μm (deeper skin layers; pooled at $5 \times 20$ μm) cryostat skin slices at different skin depths by HPLC analysis. Figure data represent mean concentration of NVX-207 at the skin depth indicated and ± SD. Data for 10-μm skin depth (*stratum corneum*) with potential test compound

residues were excluded from this figure.
(TIF)

## Acknowledgments

The authors thank Dr. Barbara Pratscher and Dr. Sabine Brandt, both Research Group Oncology, University Equine Clinic, University of Veterinary Medicine Vienna, Vienna, Austria, for providing EMM cells "eRGO1" and ES cells "sRGO1" and "sRGO2", respectively. The authors thank the Department of Pathology, University of Veterinary Medicine Hannover Foundation, Hannover, for providing equine thoracic skin for the FDC experiments. The authors thank Dr. Konstanze Bosse, Skinomics GmbH, Halle, Germany for good advice regarding questions regarding the pharmaceutical test formulations.

## Author Contributions

**Conceptualization:** Reinhard Paschke, Jessica Meißner, Manfred Kietzmann, Jessika-Maximiliane V. Cavalleri.

**Formal analysis:** Lisa Annabel Weber, Anne Funtan, Julien Delarocque.

**Funding acquisition:** Reinhard Paschke, Karsten Feige, Jessika-Maximiliane V. Cavalleri.

**Investigation:** Lisa Annabel Weber, Anne Funtan, Jutta Kalbitz.

**Methodology:** Jutta Kalbitz.

**Project administration:** Lisa Annabel Weber, Reinhard Paschke, Jessika-Maximiliane V. Cavalleri.

**Supervision:** Reinhard Paschke, Jessica Meißner, Karsten Feige, Manfred Kietzmann, Jessika-Maximiliane V. Cavalleri.

**Visualization:** Lisa Annabel Weber, Anne Funtan, Julien Delarocque.

**Writing – original draft:** Lisa Annabel Weber.

**Writing – review & editing:** Lisa Annabel Weber, Anne Funtan, Reinhard Paschke, Manfred Kietzmann.

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
