## [Decision Letter · Decision Letter 0]

9 Sep 2020

PONE-D-20-12654

In vitro assessment of triterpenoids NVX-207 and betulinyl-bis-sulfamate as a topical treatment for equine skin cancer

PLOS ONE

Dear Dr. Weber,

Thank you for submitting your manuscript to PLOS ONE. After careful consideration, we feel that it has merit but does not fully meet PLOS ONE’s publication criteria as it currently stands. Therefore, we invite you to submit a revised version of the manuscript that addresses the points raised during the review process.

Additional controls are required: normal melanocytes as non-cancerous cells; positive controls for cell assays (e.g. imiquimod). Discussion of the study limitations are necessary. High resolution figures are needed.

We look forward to receiving your revised manuscript.

Kind regards,

Irina V. Lebedeva, Ph.D.

Academic Editor

PLOS ONE

Journal Requirements:

"Some authors were funded by the Central Innovation Programme of the German Federal Ministry for Economic Affairs and Energy. LAW: Specific grant number: TopiDrugHorse 16KN051526 BMWI; AF: Specific grant number: TopiDrugHorse 16KN051530 BMWI; RP: Specific grant number: TopiDrugHorse 16KN051524 BMWI; JK: Specific grant number: TopiDrugHorse 16KN051524 BMWI. RP and JK were further funded by BioSolutions Halle GmbH (www.biosolutions-halle.de).

We note that one or more of the authors have an affiliation to the commercial funders of this research study: Biosolutions Halle GmbH,.

2.1. Please provide an amended Funding Statement declaring this commercial affiliation, as well as a statement regarding the Role of Funders in your study. If the funding organization did not play a role in the study design, data collection and analysis, decision to publish, or preparation of the manuscript and only provided financial support in the form of authors' salaries and/or research materials, please review your statements relating to the author contributions, and ensure you have specifically and accurately indicated the role(s) that these authors had in your study. You can update author roles in the Author Contributions section of the online submission form.

2.2. Please also provide an updated Competing Interests Statement declaring this commercial affiliation along with any other relevant declarations relating to employment, consultancy, patents, products in development, or marketed products, etc.  

Reviewers' comments:

Reviewer's Responses to Questions

**Comments to the Author**

1. Is the manuscript technically sound, and do the data support the conclusions?

Reviewer #1: Yes

Reviewer #2: Partly

2. Has the statistical analysis been performed appropriately and rigorously? 

Reviewer #1: I Don't Know

Reviewer #2: Yes

3. Have the authors made all data underlying the findings in their manuscript fully available?

Reviewer #1: Yes

Reviewer #2: Yes

4. Is the manuscript presented in an intelligible fashion and written in standard English?

Reviewer #1: Yes

Reviewer #2: No

5. Review Comments to the Author

Reviewer #1: In this manuscript, the authors have assessed the topical application of triterpnoids NVX-207 and betullinyl-bis sulfamate as a topical treatment for equine skin cancer. This is a follow-up in vivo study of their in vitro work published earlier.

The assays proposed in the manuscript are standard for use for such studies. The main issue if the absence of normal melanocytes as a control for the experiments with melanoma cells. The use of fibrpblasts is not an ideal control for these studies. Another concern is the resolution of figures, which makes them hard to read.

Reviewer #2: This study is a follow-on study that offers further details on the possible antitumor effects oft he triterpenoids NVX-207 and 3 betulinyl-bis-sulfamate on a variety of equine tumor cells.

Although the findings are interesting, there are some limitations to this study:

First, it would have been helpful to add a positive control to the cell assays (i.e. an anti-tumor drug for which thorough data on cytotoxicity and cell proliferation are available.

E.g. comparison to imiquimod would be interesting, as this is also used for treatment of equine sarcoids.

The general limitation is, that the MTS assay displays a cytotoxic effect for both NVX-207 and 3 betulinyl-bis-sulfamate and the index reduction of proliferation/reduction of cell viability does not look convincing for either of the substances. In addition, the cytotoxic effect on normal fibroblasts might limit its use further.

These data will only add fundamentally new insights when combined with repeated local tolerance tests (e.g. topical administration of a NVX-207 to mouse skin over a time period intended for in-vivo use in horses).

6. PLOS authors have the option to publish the peer review history of their article (what does this mean?). If published, this will include your full peer review and any attached files.

Reviewer #1: No

Reviewer #2: No

---

## [Author Response · Author response to Decision Letter 0]

29 Sep 2020

Dear Dr. Lebedeva and reviewers, 

the authors would like to thank the editor and reviewers for the review of the manuscript PONE-D-20-12654 entitled "In vitro assessment of triterpenoids NVX-207 and betulinyl-bis-sulfamate as a topical treatment for equine skin cancer". We have carefully considered your comments and our responses are given in the document entitled "Response to Reviewers".

Sincerely yours,

Lisa Weber and Jessika Cavalleri

---

## [Decision Letter · Decision Letter 1]

15 Oct 2020

In vitro assessment of triterpenoids NVX-207 and betulinyl-bis-sulfamate as a topical treatment for equine skin cancer

PONE-D-20-12654R1

Dear Dr. Weber,

We’re pleased to inform you that your manuscript has been judged scientifically suitable for publication and will be formally accepted for publication once it meets all outstanding technical requirements.

Kind regards,

Irina V. Lebedeva, Ph.D.

Academic Editor

PLOS ONE

Additional Editor Comments (optional):

Reviewers' comments:

Reviewer's Responses to Questions

**Comments to the Author**

1. If the authors have adequately addressed your comments raised in a previous round of review and you feel that this manuscript is now acceptable for publication, you may indicate that here to bypass the “Comments to the Author” section, enter your conflict of interest statement in the “Confidential to Editor” section, and submit your "Accept" recommendation.

Reviewer #2: All comments have been addressed

2. Is the manuscript technically sound, and do the data support the conclusions?

Reviewer #2: Yes

3. Has the statistical analysis been performed appropriately and rigorously? 

Reviewer #2: Yes

4. Have the authors made all data underlying the findings in their manuscript fully available?

Reviewer #2: Yes

5. Is the manuscript presented in an intelligible fashion and written in standard English?

Reviewer #2: Yes

6. Review Comments to the Author

Reviewer #2: Sorry for the "no" for Standard English, this was by accident, English is adequate!

All concerns have been adressed.

7. PLOS authors have the option to publish the peer review history of their article (what does this mean?). If published, this will include your full peer review and any attached files.

Reviewer #2: No

---

## [Editor Report · Acceptance letter]

21 Oct 2020

PONE-D-20-12654R1 

*In vitro* assessment of triterpenoids NVX-207 and betulinyl-bis-sulfamate as a topical treatment for equine skin cancer 

Dear Dr. Cavalleri:

I'm pleased to inform you that your manuscript has been deemed suitable for publication in PLOS ONE. Congratulations! Your manuscript is now with our production department. 

Kind regards, 

on behalf of

Dr. Irina V. Lebedeva 

Academic Editor

PLOS ONE